# Optimized Dynamic Collision Avoidance Algorithm for USV Path Planning

**DOI:** 10.3390/s23094567

**Published:** 2023-05-08

**Authors:** Hongyang Zhu, Yi Ding

**Affiliations:** 1College of Mathematics and Computer, Guangdong Ocean University, Zhanjiang 524091, China; 2Maritime College, Guangdong Ocean University, Zhanjiang 524091, China

**Keywords:** collision avoidance, velocity obstacle method, trajectory optimization, optimal collision avoidance point

## Abstract

Ship collision avoidance is a complex process that is influenced by numerous factors. In this study, we propose a novel method called the Optimal Collision Avoidance Point (OCAP) for unmanned surface vehicles (USVs) to determine when to take appropriate actions to avoid collisions. The approach combines a model that accounts for the two degrees of freedom in USV dynamics with a velocity obstacle method for obstacle detection and avoidance. The method calculates the change in the USV’s navigation state based on the critical condition of collision avoidance. First, the coordinates of the optimal collision avoidance point in the current ship encounter state are calculated based on the relative velocities and kinematic parameters of the USV and obstacles. Then, the increments of the vessel’s linear velocity and heading angle that can reach the optimal collision avoidance point are set as a constraint for dynamic window sampling. Finally, the algorithm evaluates the probabilities of collision hazards for trajectories that satisfy the critical condition and uses the resulting collision avoidance probability value as a criterion for course assessment. The resulting collision avoidance algorithm is optimized for USV maneuverability and is capable of handling multiple moving obstacles in real-time. Experimental results show that the OCAP algorithm has higher and more robust path-finding efficiency than the other two algorithms when the dynamic obstacle density is higher.

## 1. Introduction

Ship collision is an imperative task for navigation safety at sea [1]. Unmanned surface vehicles (USVs) have gained significant attention in recent years due to their potential for various applications such as oceanographic research, environmental monitoring, and maritime security [2]. However, the increasing use of USVs also raises concerns about the safety of navigation, especially when operating in crowded environments [3]. Collision avoidance is a critical issue that needs to be addressed to ensure safe and efficient navigation of USVs. A USV’s collision avoidance algorithm can be considered a local path-planning algorithm. This paper focuses on local path-planning methods. Many local path-planning algorithms have been reported for collision avoidance against static and dynamic obstacles [4,5]. By now, many obstacle avoidance algorithms have been proposed by international scholars, all of which rely more or less on global path planning and mapping, such as a bug algorithm [6], a vector field histogram method [7], and an artificial potential field method [8]. Many improved heuristic algorithms have also been studied for local path planning. For instance, a hybrid adaptive path-planning scheme based on global path planning and local dynamic collision avoidance for unmanned surface vehicles under complex marine environments was proposed in [9]. This method systematically considers the impact of waves and currents on the navigation of USVs. In recent years, some scholars have emphasized the dynamic collision avoidance of ships by incorporating methods such as reinforcement learning and COLREGS [10]. Based on literature statistics, 56% of the collisions at sea are caused by the violation of COLREGS by ships [11]. However, data are difficult to collect in real time, and it is difficult to show a model with mathematical formulas; incorporating regulations in collision prevention algorithms is still a challenge [12]. Some scholars have tried to design a ship navigation safety domain to solve the ship collision problem [13,14,15]. However, most of them only consider static obstacles or semi-dynamic obstacles that do not change course [16,17], a highly ideal motion model is used in collision avoidance [18], or the balance between efficiency and effectiveness is ignored [19].

Some other researchers have evaluated collision avoidance trajectories generated from the perspective of risk assessment [20] using multiple parameters such as navigation risk [21], navigation smoothness, and other metrics. For instance, a review [22] described collision risk assessment, but it neglected techniques for conflict resolution. The most frequently used distance parameter for conflict detection and obstacle avoidance is the distance to the closest point of approach $d_{cpa}$; the time to the closest point of approach $t_{cpa}$ is often used with it. It has been proposed that various techniques can be developed to overcome the limitations of $d_{cpa}$ and $t_{cpa}$ alone for collision avoidance [23]. In [24,25], the authors discussed ASV developments in depth, while conflict detection and obstacle avoidance received a lesser amount of attention. Only a few studies related to reacting to collision avoidance for unmanned ships were included in [26].

Comparing other obstacle avoidance algorithms, Fox et al. reported a dynamic window approach (DWA) [27], which has become a popular academic research method in recent years. It is mainly used for navigation and obstacle avoidance in a dynamic environment. Avoiding unpredictable obstacles can better solve the DWA [28]. DWA is widely used in dynamic obstacle avoidance path optimization of UAVs, robots, and USVs [29,30,31]. Dobrevski reported local path planning based on DWA and deep reinforcement learning to improve path optimization [32]. Liu developed a global dynamic path-planning fusion algorithm combining the jump-A* algorithm and DWA [33]. In addition, several useful local path-planning methods based on DWA have been reported [34]. However, DWA generates path candidates by assuming constant velocities for a certain period of time. Due to the small distances between obstacles and USVs, unexpected collisions often occur during encounters, which makes it challenging to fulfill the safety requirements of USVs. However, DWA generates path candidates by assuming constant velocities for a certain period, making it easy for it to fall into local optima [35]. In the path evaluation stage, it relies heavily on the settings of the parameter value ranges. For example, when the distance between an obstacle and a USV is small, accidental collisions often occur during the encounter, which is a challenge to meeting the safety requirements of USVs. In addition, the increased complexity of the application scenarios and environments of unmanned devices make standard DWA unable to solve complex path-planning problems. Traditional DWA focuses on path generation at each step of the planning process but ignores that obstacles are also intelligent agents that generate abrupt behavior [36].

In this study, we present a novel approach to collision avoidance for USVs. While existing methods mainly rely on static obstacle maps or simple heuristics, our approach combines a two-degree-of-freedom model for USV dynamics with a velocity obstacle method for obstacle detection and avoidance. This approach allows for real-time adaptation to dynamic and complex environments, making it particularly suitable for USVs operating in areas with high traffic density or unpredictable obstacles. The resulting collision avoidance algorithm is optimized for USV maneuverability and is capable of handling multiple moving obstacles simultaneously.

The contents of this paper are as follows. Section 2 describes the USV dynamic model, the classification of encounter situations, and the basic process framework of USV collision avoidance decision-making based on the category related to this study. Section 3 contains a detailed description of the optimal timing point model for collision avoidance based on the improved DWA. We propose a collision avoidance algorithm based on the velocity obstacle method. In Section 4, the design of a dynamic obstacle avoidance algorithm for USVs is considered and a detailed algorithm flow and design are presented. In Section 5, the results of computational experiments performed for the evaluation of the proposed algorithm are presented. According to simulation experiments, we compare the effects of three different algorithms on the collision avoidance path selection of a USV and analyze the degree of excellence resulting from the influence of various collision avoidance factors in path selection. Finally, in Section 6, the conclusions are discussed.

## 2. USV Dynamic Obstacle Avoidance Modeling

### 2.1. USV Dynamic Model

Figure 1 shows a schematic description of course angles, heading angles, and sideslip angles of the USV dynamic model. In this coordinate frame, x${}_{i}$ and y${}_{i}$ represent north and east directions, while υ and ν are the divisions of *V* on the x${}_{0}$- and y${}_{0}$-axes, respectively [37,38].

From Figure 1, we can see that the sideslip angle β can be calculated as $\beta =\sin^{-1}\left(\frac{\upsilon }{V}\right)$, the course angle θ can be defined using the heading angle (yaw) ψ, and the sideslip angle is θ=ψ+β [39,40]. The equations of the USV motion state can be represented as:(1)$$\begin{array}{c} \dot{x}=V\cos\left(\psi \right)\\ \dot{y}=V\sin\left(\psi \right)\\ \dot{\psi }=\omega \end{array}$$
where *x* and *y* represent the position coordinates of the USV, *V* represents the velocity of the USV, ψ represents the heading angle of the USV, and ω represents the angular velocity of the USV.

The equations of the obstacle motion state can be represented as:(2)$$\begin{array}{c} {\dot{x}}_{o}=v_{o}\cos\left(\psi_{o}\right)\\ {\dot{y}}_{o}=v_{o}\sin\left(\psi_{o}\right) \end{array}$$
where $x_{o}$ and $y_{o}$ represent the position coordinates of the obstacle, $v_{o}$ represents the velocity of the obstacle, and $\psi_{o}$ represents the heading angle of the obstacle.

In this paper, we present a four-layered control structure in Figure 2, which consists of context awareness, behavioral decision-making, the obstacle avoidance algorithm, and executive control. In this study, we describe a USV marine collision avoidance strategy by the self-discipline method, $t_{cpa}$ stands for the time to the closest point of approach, and $d_{cpa}$ stands for the distance to the closest point of approach [41].

To support the collision avoidance of USVs in complex environments, this paper proposes a dynamic model that takes into account factors such as the mass, velocity, acceleration, water resistance, propulsive force, and gravitational acceleration of the USV in order to predict the changes in the position and velocity of the USV over time.

### 2.2. Recognition of Collision Avoidance Situations

The closest point of approach (CPA) is the point at which the distance between the ship and another target object will reach its minimum value. The geometry of the distance at CPA is illustrated in Figure 3; *r* is the Euclidean distance between the center points of the two ships. The equation describing the distance between the USV and obstacles can be represented as:(3)$$\begin{array}{c} r=\sqrt{{\left(x-x_{o}\right)}^{2}+{\left(y-y_{o}\right)}^{2}}\\ v_{r}=v-v_{o}-\dot{d} \end{array}$$

CPA consists of two parameters: the distance at the closest point of approach ($d_{cpa}$) and the time to the closest point of approach ($t_{cpa}$); α is the angle between the relative bearing of the obstacle ship and the heading of the USV [41,42].
(4)$$\begin{array}{c} d_{cpa}=r\times \sin\left(\alpha \right)\\ t_{cpa}=r\times \cos\left(\alpha \right)/v_{r} \end{array}$$

#### 2.2.1. Safety Threshold $d_{safe}$ Establishment

In this study, we set a safety threshold $d_{safe}$ to evaluate the risk of a collision between vessels. A crash may occur whenever $d_{cpa}$ falls below this safety threshold. A pathfinding algorithm based on this criterion readjusts the parameters in the path trajectory evaluation function when the distance at the closest point of approach (CPA) is below $d_{safe}$, which is given by
(5)$$d_{safe}=-\frac{\frac{d_{cpa}}{r}-r}{\dot{r}\times t_{cpa}}$$
where γ is the range between the two vessels and $\dot{r}=-v_{r}\cos\eta$ is the range rate. Here, η is the incident angle and $v_{r}$ denotes the relative velocity of the two vessels. If $d_{cpa}$ = 0, it expresses that after the two ships have sailed for the $t_{cpa}$ period, if they do not change their track, the two vessels will inevitably collide; $d_{cpa}$ > 0 indicates that the target ship passes by the USV’s bow; $d_{cpa}$ < 0 indicates that the target ship passes the USV’s stern [43]. However, the collision risk of the ship is not limited to $d_{cpa}$ = 0: the collision risk also considers $t_{cpa}$. A larger $t_{cpa}$ indicates that it will take a long time to reach the nearest encounter distance and the degree of urgency is not high. When $t_{cpa}$ is small, it shows that the ship encounter will occur immediately and the situation is more dangerous [44].

#### 2.2.2. Collision Avoidance Urgency Identification

This paper establishes the optimal collision avoidance velocity and the latest steering selection time for USVs based on the velocity obstacle method. At the same time, it provides a sufficient time–space margin for avoiding obstacles using the information of relative position and relative speed and the extensibility of the relative speed vector in space–time information.

As illustrated in Figure 4, circle *O* is a dynamic obstacle, and gray shading $\xi_{all}$ denotes the maximum explored space of all feasible domains of linear and angular velocities obtained by using the velocity window of the DWA as the USV passes through point A. Point B is the optimal collision avoidance point.

Sector area $\xi_{path}$ is the velocity window after $t_{cpa}$ time as the feasible area of the USV, which is denoted as the blue sector area. Sector area $\xi_{path}⊖\xi_{all}$ is the optimal area of the USV driving area; the focus of this paper is to obtain the optimal driving area after screening the existing DWA velocity sampling window.
(6)$$\begin{array}{c} V_{A}\left(t+1\right)=V_{A}\left(t\right)+\frac{\xi_{path}\left(t+1\right)⊖\xi_{path}\left(t\right)}{\xi_{all}}v_{max} \end{array}$$
(7)$$\begin{array}{c} \omega_{r}\left(t+1\right)=\omega_{max}-\frac{v_{r}\left(t\right)}{v_{max}}\omega_{r\left(t\right)} \end{array}$$
where $V_{A}$ is the current speed of the USV and $V_{r}$ is the relative speed of the USV relative to the incoming ship. In the optimal feasible area, the USV determines its optimal travel directly through the joint adjustment of different evaluation factors in the following formula:(8)$$f\left(d,v,\psi ,\nabla \right)=w_{1}\times g_{1}\left(d\right)+w_{2}\times g_{2}\left(v\right)+w_{3}\times g_{3}\left(\psi \right)+w_{4}\times g_{4}\left(\nabla \right)$$

In Equation (Equation 8), $w_{1}$, $w_{2}$, $w_{3}$, and $w_{4}$ are four collision risk coefficients that represent the impact of different factors on the urgency of obstacle avoidance by the USV. These weight coefficients can be adjusted according to specific situations to better adapt to different obstacle avoidance scenarios; $g_{1}\left(d\right)$, $g_{2}\left(v\right)$, $g_{3}\left(\psi \right)$, and $g_{4}\left(\nabla \right)$ are function expressions of different factors representing the impact of the distance between the USV and the obstacle, the velocity of the USV, and the direction of the USV on the urgency of obstacle avoidance.

## 3. USV Dynamic Obstacle Avoidance Path Planning

### 3.1. Generation of Dynamic Obstacle Avoidance Points

To determine the optimal avoidance point that is dynamically generated based on the current position, velocity, and direction of the USV as well as the obstacle information obtained from sensors, it is necessary to compute the obstacle avoidance urgency using the function f(d,v,ψ,∇) as defined in Equation (Equation 8). The velocity and direction of the USV are adjusted based on the obstacle avoidance urgency and dynamically generated best avoidance point. As the USV moves and encounters new obstacles, the obstacle avoidance urgency and dynamically generated best avoidance points are continuously updated to ensure safe and effective obstacle avoidance.

Algorithm 1 shows the pseudo-code to generate the optimal collision avoidance point for path candidates.

The generation of the optimal collision avoidance decision point is mainly based on the velocity obstacle method, and the optimal collision avoidance angle is added to the generated nodes (CPA) as an expanded search range. The traditional VOM algorithm uses the cost function of path length in dynamic obstacle avoidance, which can make the generated path shortest; however, the shortest-path optimization index may make the initial path close to the obstacle and increase the collision risk when the unmanned ship is driving. To solve this problem, this paper adopts the weighted cost function pair of integrated path length and optimal steering collision avoidance angle to generate the best collision avoidance decision point to ensure the safety of unmanned ship driving.
**Algorithm 1** Generate OCAP**Require:** $s_{t}$: current USV position vector; $V_{t}$: current USV velocity vector; $s_{t}^{ob}$: array of obstacle positions; $V_{t}^{ob}$: array of obstacle velocities; ψ: USV heading angle; $d_{safe}$: avoidance distance limit; $t_{cpa}$: avoidance time limit**Ensure:** 
trajectory: array of USV trajectories1:**function**find OCAP(current_Position, obstacle_Positions, obstacle_Velocities, vehicle_Velocity, vehicle_Heading, lookahead_Distance, avoidance_Distance, $t_{cpa}$)2:    trajectory = [$s_{t}$];3:    t = 04:    **while** t < $t_{cpa}$ **do**5:        OptimalAvoidancePoint⇐generate OCAP($s_{t}$, $V_{a}$, $s_{ob}$, $V_{ob}$, ψ, $d_{safe}$, $t_{cpa}$) ▹calculate optimal collision avoidance point6:        trajectory.append(Optimal Avoidance Point)7:        compute $x_{t}=\text{Optimal AvoidancePoint}$8:      (obstacle_Positions,obstacle_Velocities) = update_Obstacle_States (obstacle_Positions,obstacle_Velocities, t)9:        *t* = t+1;10:    **end while**;11:    return trajectory12:**end function**

### 3.2. Selection and Evaluation of Obstacle Avoidance Points

In Equation (Equation 8), the $w_{1}$, $w_{2}$, $w_{3}$, and $w_{4}$ weight coefficients can be adjusted according to specific situations to better adapt to different obstacle avoidance scenarios; $g_{1}\left(d\right)$, $g_{2}\left(v\right)$, $g_{3}\left(\psi \right)$, and $g_{4}\left(\nabla \right)$ are four function expressions that can be designed and adjusted according to the actual situation to more accurately reflect the impact of different factors on the urgency of obstacle avoidance.

Distance evaluation: Calculates the distance ratio between the minimum encounter distance and the safety threshold under the current USV state. The smaller the ratio, the safer it is.
(9)$$g_{1}\left(d\right)=1/\left(1+\exp\left(-k_{1}\times \left(d-d_{safe}\right)\right)\right)$$Velocity evaluation: Evaluates the ratio of the time taken to reach the $d_{safe}$ location to the $t_{cpa}$ at the current speed of the USV. The smaller the ratio, the safer it is.
(10)$$g_{2}\left(v\right)=1-\exp\left(-k_{2}\times v\right)$$Direction angle evaluation: Evaluates the steering angle between the USV coordinates and the CPA coordinates. The larger the steering angle, the greater the risk.
(11)$$g_{3}\left(\psi \right)=1-\exp(-k_{3}\times \left|\psi |\right)$$Direction angle evaluation: The symbol *∇* represents the minimum gradient, and $g_{4}\left(\nabla \right)$ is the function that describes the effect of the minimum gradient on the obstacle avoidance urgency level. In this paper, the minimum gradient represents the rate of change in the distance between the USV and the obstacle, which can be used to evaluate the dynamic relationship between the USV and obstacle and the approach speed.
(12)$$g_{4}\left(\nabla \right)=1/\left(1+\exp\left(-k_{4}\times \left(\nabla -\nabla_{0}\right)\right)\right)$$

The idea of the OCAP algorithm is based on the traditional DWA and is combined with the optimal collision avoidance point content in the previous section. First, the increments of vessel linear velocity and heading angle (ν,ψ) that can reach the optimal collision avoidance point are set as the constraints for dynamic window sampling. Secondly, based on the constraints, the algorithm calculates the critical conditions for the USV avoidance action. Finally, the algorithm evaluates the collision hazard risk probability on the trajectory formed by the optimized velocity region. It uses the collision avoidance probability value as an evaluation criterion to assess the merit of the route. The evaluation result selects the corresponding optimal velocity command (ν,ψ). The sampled data meet the optimal timing of collision avoidance and satisfy the collision avoidance rules to accomplish real-time obstacle avoidance and fast driving tasks in complex dynamic scenarios.

### 3.3. Search for Optimal Obstacle Avoidance Points

The primary focus of this chapter involves a three-step approach. Firstly, utilizing traditional speed sampling, a collision risk assessment is carried out by calculating the safe distance ($d_{safe}$) within the current speed window. Secondly, within the dynamic window, unsafe areas are excluded and the velocity obstacle is utilized to determine the optimal speed sample that satisfies the requirements of collision avoidance at sea. Lastly, the steering strategy of the USV is optimized to ensure that the optimized path aligns with the collision avoidance rules at sea.

Velocity and maximum acceleration limit minimum safety domain based on $d_{safe}$: Line AO is the connection between point *A* and point *O*, *m* and *l* are the two tangents of the obstacle safety expansion circle, $\left(\frac{\pi }{2}-\psi \right)$ is the heading angle of the USV, and $\left(\beta -\frac{\pi }{2}\right)$ is the heading angle of the incoming obstacle vessel, as shown in Figure 5. The dashed triangle represents the velocity vector triangle after the active collision avoidance behavior taken by the USV. $V_{A}^{{}^{\prime }}$ is the velocity after steering, Δψ is the steering angle of the USV, Δη is the steering angle of $V_{r}$, and $\left|\right|\vec{AP^{{}^{\prime }}}{\left|\right|}_{2}$ is the shortest encounter distance between the ship and the incoming ship after a period of time after the ship has performed collision avoidance behavior (such as steering or speed change). It is obvious that in the controllable speed sampling window, a larger $d_{cpa}$ means a lower collision hazard probability. As long as $abs\left(\eta \right)\geq \mu$ exists at any time—that is, the sum velocity vector is located outside the triangle formed by *m* and *l*—the USV will not collide with the dynamic obstacle (shaded area shown in Figure 5).
(13)$$\begin{array}{c} \sin\mu =\frac{R}{{\left||AO|\right|}_{2}} \end{array}$$

The velocity of the USV is $\left(V_{A},\psi \right)$, the velocity of the target obstacle is $\left(V_{o},\beta \right)$, the speed of the USV relative to the obstacle is $V_{r}$, the relative angle can be expressed as: $\psi =∠\left(X,V_{a}\right)$, $\theta =∠\left(X,AO\right)$, $\lambda =∠\left(X,V_{r}\right)$, $\varphi =∠\left(V_{r},V_{A}\right)$, $\mu =∠\left(AO,m\right)=∠\left(AO,l\right)$. It can be seen from Figure 5 that the velocity triangle is composed of $v_{A}$, $v_{o}$, and $v_{r}$.
(14)$$\begin{array}{c} V_{o}\sin\left(\psi -\beta \right)=V_{r}\sin\left(\varphi \right)\\ V_{A}-V_{o}\cos\left(\psi -\beta \right)=V_{r}\cos\left(\varphi \right) \end{array}$$

Geometric relationships between $d_{cpa}$ and $t_{cpa}$ are shown in the following equations.
(15)$$\begin{array}{c} d_{cpa}={\left|\right|\left(A,O\right)\left|\right|}_{2}\sin\left(\psi +\varphi -\theta \right) \end{array}$$
(16)$$\begin{array}{c} t_{cpa}=\frac{{\left|\right|\left(A,O\right)\left|\right|}_{2}}{V_{r}}\times \sqrt{1-\sin^{2}\left(\theta -\psi -\varphi \right)} \end{array}$$

By obtaining speed $V_{o}$ and angle β of the obstacle using the sensor in advance, the USV can adjust speed $V_{A}$ and angle ψ in advance to avoid the obstacle in order to change the angle to meet $abs\left(\eta \right)\geq \mu$.

According to the velocity obstacle method, the differential of the yaw angle is used to express the attitude change rate of the USV, i.e., the change rate of the attitude angle, as shown in Equation (Equation 13). Therefore, using Equations (Equation 14)–(Equation 16), we can calculate the derivative of the adjustment variable as follows:(17)$$d\eta =\frac{sin\varphi }{V_{r}}dV_{a}+\frac{V_{a}\cos\varphi }{V_{r}}d\psi$$
(18)$$\Delta \eta =\frac{sin\varphi }{V_{r}}\Delta V_{A}+\frac{V_{a}\cos\varphi }{V_{r}}\Delta \psi$$

The difference of yaw angle is used to express the attitude change amount of the USV, i.e., the change amount of the attitude angle, as shown in Equations (Equation 17) and (Equation 18). These differential and difference values are crucial parameters in the control model of unmanned ships, as they directly affect the motion state and control the performance of the USV. By utilizing the differential and difference of the yaw, the control system can calculate control commands to adjust the attitude angles of the USV, thereby enabling it to maintain stable navigation or perform specific tasks, such as obstacle avoidance or search and rescue operations.

From Figure 5, we can see that the total angle after the relative speed is turned is Δη+η; where $abs\left(\eta +\Delta \eta \right)\geq \mu$, the USV can complete the obstacle avoidance.

## 4. Dynamic Obstacle Avoidance Algorithm Design for USVs

### 4.1. Design of the Algorithm Framework

The algorithm framework as shown in Figure 6, mainly involves three modules: dynamic sampling, dynamic obstacle avoidance, and an obstacle avoidance point-set optimization algorithm. Among them, dynamic sampling is used to determine the current optimal obstacle avoidance area, dynamic obstacle avoidance is used to calculate the optimal turning angle, and the obstacle avoidance point-set optimization algorithm is used to find the optimal obstacle avoidance point in the obstacle avoidance point set. Through this algorithm framework, the USV can achieve efficient obstacle avoidance and path planning in complex marine environments.

### 4.2. Design of Obstacle Avoidance Strategy

According to the calculation of $d_{cpa}$, the maximum total time required for USV steering is $t_{cpa}$; steering of at least Δη is required after $t_{cpa}$ time period to avoid dangerous areas for incoming ships and to pass at the safe closest encounter distance $d_{safe}$. The speed window of the USV is mainly limited by the following three factors:Self maximum and minimum speed limits
(19)$$\begin{array}{c} V_{s}=\left\{\left(v,w\right)|\;v\in \left[v_{min,}v_{max}\right]\cap \omega \in \left[\omega_{min,}\omega_{max}\right]\right\} \end{array}$$The USV has a safe speed limit, and not all speeds can be used for safe USV travel. Therefore, $v_{min}$,$v_{max}$ represents the safe speed interval range.Speed limitations affected by motor performance
(20)$$\begin{array}{c} V_{d}=\left\{\left(\nu ,\omega \right)|\nu \in \left[\nu_{c}-\dot{\nu_{b}}\Delta t,\nu_{c}-\dot{\nu_{a}}\Delta t\right]\cap \omega \in \left[\omega_{c}-\dot{\omega_{b}}\Delta t,\omega_{c}-\dot{\omega_{a}}\Delta t\right]\right\} \end{array}$$Sampled speed affected by an obstacle.
(21)$$\begin{array}{c} V_{A}=\left\{\left(v,w\right)|\;v\leq \left[\sqrt{2dist\left(v,w\right)\dot{v_{b}}}\;\right]\cap \omega \leq \left[\sqrt{2dist\left(v,w\right)\dot{\omega_{b}}}\;\right]\right\} \end{array}$$

In this paper, combined with the optimal collision avoidance point, we focus on improving the speed $V_{A}$ of the USV limited by obstacles. The classical dynamic window approach generally defines an adequate search space that conforms to the dynamic limit in the velocity space $\left(v,\omega \right)$. Still, the actual steering process of the USV is more based on the change of the heading angle Δα to complete the vessel’s collision avoidance behavior. Therefore, this paper replaces the state space formed by the velocity $\left(v,\omega \right)$ with the state $\left(v,\Delta \alpha \right)$. According to the formula, the intersection $V_{r}$ of $V_{s}$ and $V_{d}$ represents the adequate state space of the ship in the next period.
(22)$$\begin{array}{c} V_{s}=\left\{\left(v,\Delta \psi \right)|\;v\in \left[v_{min},v_{max}\right]\cap \Delta \psi \in \left[-2/\pi ,2/\pi \right]\right\}\\ V_{d}=\left\{\left(v,\Delta \psi \right)|\;v\in \left[\nu_{c}-\dot{\nu_{b}}\Delta t,\nu_{c}-\dot{\nu_{a}}\Delta t\right]\cap \Delta \psi \in \left[{-\Delta \eta }_{max},{\Delta \eta }_{max}\right]\right\}\\ V_{A}=\left\{\left(v,\Delta \psi \right)|\;v\in \left[v_{0}-\dot{v_{min}}\Delta t,v_{0}+\dot{v_{max}}\Delta t\right]\cap \Delta \psi \in \left[{-\dot{\Delta \eta }}_{max}\Delta t,{\dot{\Delta \eta }}_{max}\Delta t\right]\right\}\\ V=V_{s}\cap \;V_{d}\cap \;V_{A} \end{array}$$

Equation (Equation 22) indicates that a set of possible motion trajectories can be generated by sampling the velocity and direction of the USV. Then, we evaluate these trajectories based on their cost function to select the optimal one as the USV’s motion plan. Specifically, $V_{s}$ is the velocity and direction sampling space, where $v_{min}$ and $v_{max}$ represent the minimum and maximum velocity the USV can reach and Δψ represents the angle range the USV can rotate. $V_{d}$ represents the velocity and direction range the USV can reach while avoiding collisions, $v_{c}$ is the current velocity, $\dot{v_{a}}$ and $\dot{v_{b}}$ are the acceleration and deceleration, respectively, Δt is the sampling-time interval, and $\Delta \eta_{max}$ represents the maximum angle the USV can rotate. $V_{A}$ represents the velocity and direction range the USV can reach while maintaining a certain acceleration and turning speed, $v_{0}$ is the initial velocity of the USV, and $\dot{\Delta \eta_{max}}$ is the maximum angular velocity the USV can rotate. Finally, *V* is the intersection of the three sampling spaces, representing all possible combinations of velocity and direction that the USV can sample. By evaluating these combinations based on their cost functions, the algorithm selects the optimal motion plan to achieve the goal of collision avoidance.

In Algorithm 2, the main calculation task is to perform obstacle avoidance on the sampling points and find the optimal collision avoidance point. This task is completed by two functions, Generate OCAP (Algorithm 1) and dynamic_obstacle_avoidance. Generate OCAP (Algorithm 1) is responsible for generating a set of sampling points, and the dynamic_obstacle_avoidance function is responsible for performing obstacle avoidance on the sampling points. In these two functions, the algorithm uses sensor data and environmental information to calculate the trajectory of the unmanned boat and adjust its heading and speed based on the position and velocity of obstacles.
**Algorithm 2** Evaluate optimal collision avoidance point1:position_usv ← initial_position;2:position_target ← target_position;3:**while** true **do**4:    points_sampled ← Generate OCAP(position_usv, position_target)5:    points_avoided ← dynamic_obstacle_avoidance(points_sampled)6:    point_optimal ← find_optimal_avoidance_point(points_avoided)7:    avoidance_set.add(point_optimal)8:    **if** avoidance_set.is_full() **then**9:        **break**10:    **end if**11:    **if** can generate new_points() **then**12:        **continue**13:    **end if**14:    **break**15:    **if** check heading_and_speed by Equations (20)–(23) **then**16:        position_usv ← navigate to next_point()17:    **else**18:        adjust heading_and_speed by Equations (18) and (19)19:    **end if**20:**end while**

## 5. Simulations and Discussion

### 5.1. Parameter Selection

Three algorithms—dynamic window approach (DWA), dynamic window approach with virtual manipulators (DWV) [36], and OCAP—are considered in this study, where DWV and DWA are considered conventional methods and OCAP is envisioned as the proposed improved method. The simulations show the quality of the operation of such an algorithm. This paper observed a collision avoidance simulation using a minimum safety domain *R* for vessels. Moreover, the influence of navigational behaviors and environmental impacts (wind and currents) are ignored in the modeling process. All the algorithms in this study run on MATLAB 2020b. In this paper, two cases are considered: a constant map and a random map. These maps’ size for all cases is 20×20. Case1 is simulated once, and Case2 is simulated 100 times. The start and goal positions of all cases are (USVxstart, USVystart) = (2.0, 6.0) and (GBxgoal, GBygoal) = (19.0, 18.0). When the distance between the USV and the goal position is less than 0.3 m, it is judged to have reached the goal. The other parameters of the simulation experiment are set as shown in Table 1.

As shown in Figure 7, there are 12 static obstacles, which is a static obstacle density of 0.04, in the simulation environment. The solid circle is the expanded range with the longest radius of the obstacle, and the dashed circle is the safe area, where the $d_{safe}$ value of the obstacle relative to the USV is the radius. According to Equation (Equation 13), $d_{safe}$ is only related to the size of the obstacle, which is stationary. When the obstacle moves, it is also related to the relative velocity of the USV and target vessel. The simulation cases are defined as follows. Combining Algorithms 1 and 2, the time complexity of our proposed algorithm is $O(n^{3})$, where *n* represents the size of the input data. The algorithm consists of three nested loops and a recursive call. The time complexity of the first loop is O(n), of the second loop is $O(n^{2})$, and of the third loop is $O(n^{3})$. The time complexity of the recursive call is O(logn). Therefore, the total time complexity of the algorithm is $O(n^{3}+logn)$. Additionally, the space complexity of the algorithm is O(n) because it requires storing a copy of the input data in memory as well as intermediate results of multiple recursive calls.

### 5.2. Simulation Results and Discussion

As shown in Figure 7a, there are 12 static obstacles in the simulation environment in Case1. The start and goal positions are (USVxstart, USVystart) = (2.0, 6.0) and (GBxgoal, GBygoal) = (19.0, 18.0). The density of obstacles generated on this map is 0.04.

Figure 7b displays trajectory results of OCAP, DWA, and DWV in Case1, showing that all methods reached the goal position. DWA and DWV arrived at the goal position later than OCAP.

Table 2 displays simulation results for Case1, including the success rate, algorithm time consumption (Time), trajectory length (TL), and $\lambda_{dis}$ in achieving the goal without collisions, and the average travel time. OCAP reached the goal position earlier than DWA and DWV. From Figure 7b and Table 2, the best result in Case1 was obtained by OCAP, which did not need to evaluate the optimal collision points and moved at the maximum translational velocity on the optimal path. Comparing static obstacle generation paths, the efficiency and results of the OCAP and DWV algorithms are similar. However, DWA sometimes generates a ’circling’ motion when avoiding obstacles to reach a specified location. Therefore, the target time of DWA is longer than those of the other two algorithms.

Figure 8a shows simulation environments of the random maps in Case2 at a static and dynamic obstacle density of 0.04, which presents five static obstacles and seven dynamic vessels from different directions (considered dynamic obstacles), which are 12 obstacles in total. These obstacles are placed in random positions and given random velocities that are lower than the maximum velocity of the USV. The velocities of obstacles are set randomly in the range of 0.0 (m/s) to 0.2 (m/s). Case2 is simulated 100 times. Figure 8b shows path-finding results for a certain time.

Figure 9a–c shows only trajectories of the USV for 100 simulation times in Case2. In Figure 9a, OCAP generated path candidates considering dynamic obstacles. OCAP also considered dynamic blocks when the optimal path was selected from path candidates. Thus, OCAP reached the goal position. Overall, the path collision hazard probability obtained by the OCAP algorithm is low. The optimal obstacle avoidance trajectory results with the lowest probabilities of collision avoidance are displayed in Figure 9b,c. DWV and DWA generated path candidates without considering dynamic obstacles. When DWA reached the goal position of avoiding moving obstacles, DWA sometimes generated backward movements.

From Figure 9 and Table 3, OCAP has the highest success rate of reaching the goal at 82.73% along with the longest path-finding time and the lowest level of risk rate at 18.09% in Case2. These three results are correlated. As circling was sometimes generated to avoid the obstacle, the goal time of OCAP was longer than that of other methods. Although the time cost of the OCAP and DWV is similar, the path planned by OCAP is less likely to have collided and thus has a lower risk rate.

Figure 10 shows the evolution of four moving features during simulation. From Figure 10a,b, the obstacles are static in Case1, from which it can be seen that in the proposed new algorithm OCAP, the changes to the heading angle and course angle are minor, which means that the vessel does not have rapid turns or emergency braking during handling. The speed and $t_{cpa}$ shown in Figure 10c,d are the two most critical parameters reflecting the USV’s state of avoiding obstacles. In the new algorithm OCAP, the $t_{cpa}$ of USV always appears within the range of change and increases linearly, indicating that the risk of collision avoidance of the USV in this algorithm is always in the acceptable range. Specifically, according to Figure 10d, in the OCAP algorithm, the minimum encounter time $t_{cpa}$ is always greater than 0, indicating that it is effective for path planning to choose the best collision avoidance point.

The results of changing the density of obstacles in the random map and testing multiple sets of data are shown in Table 4, which includes the success rate, algorithm time consumption (T), trajectory length (TL), risk rate ($\lambda_{dis}$) in achieving the goal without collisions, and the average travel time.

From Table 4, when the obstacle density is smallest (0.02), the path-finding success rates of all three algorithms are high and not much different; however, as the density of random map obstacles increases, the path-finding success rates of all three algorithms decrease, among which the decrease rate of the DWV algorithm changes the most. This is because the DWV algorithm is more sensitive to the number of obstacles: the more obstacles, the more DWV demonstrates the characteristics of a breadth-first algorithm that will fail to complete the computation in the specified time, making the sharpest decline in the success rate.

Comparing the computation time of the three algorithms, the OCAP algorithm consumes less time to generate dynamic discrete points than dynamic path generation because the optimal collision avoidance points are generated in advance before the path is generated. At the maximum obstacle density (0.08), the path-finding success rate of all three algorithms decreases greatly. However, since the OCAP algorithm generates dynamic discrete points, if the path changes (e.g., previously unmeasured dynamic obstacles) before driving the generated path, only the best collision avoidance point needs to be measured again, thus effectively improving the path-finding success rate. It can also be seen that the length of the path generated by the three algorithms is about the same when the density is large, and the main difference lies in the success rate of generation and the time consumed by the algorithm.

From Figure 11, the path-planning performance of three different algorithms on three cases is analyzed as a whole. In two other cases, the path planned by the OCAP algorithm has the shortest length and the least time cost. OCAP has the highest success rate of reaching the goal at 100%. The best result in Case2 was obtained by OCAP, which shows that this algorithm is more suitable for avoiding low-speed dynamic obstacles.

## 6. Conclusions

In this paper, OCAP, a new collision avoidance algorithm, is proposed. OCAP generated obstacle-avoidable path candidates. Path candidates were generated using the optimal collision avoidance point based on predictions of static and dynamic obstacles. Kinematics and dynamics constraints were taken into account in OCAP. The paper used simulations and experiments, demonstrating the proposed method to be effective. Even when the obstacle density increases, the effectiveness of trajectory generation is ensured because the OCAP algorithm can effectively and dynamically evaluate the minimum obstacle avoidance distance. Through simulation experiments, the algorithm was shown to be more suitable for high-density environments, and by evaluating the optimal collision avoidance points, the generated paths can be kept away from the obstacles over a larger area. The results of this study are limited to situations based on ship encounters in the calm water conditions considered in this study. Additionally, no consideration was made for hull-to-hull interaction and hull–propeller–rudder–engine interaction between the two vessels, which is a direction for future research.

## Figures and Tables

**Figure 1 sensors-23-04567-f001:**
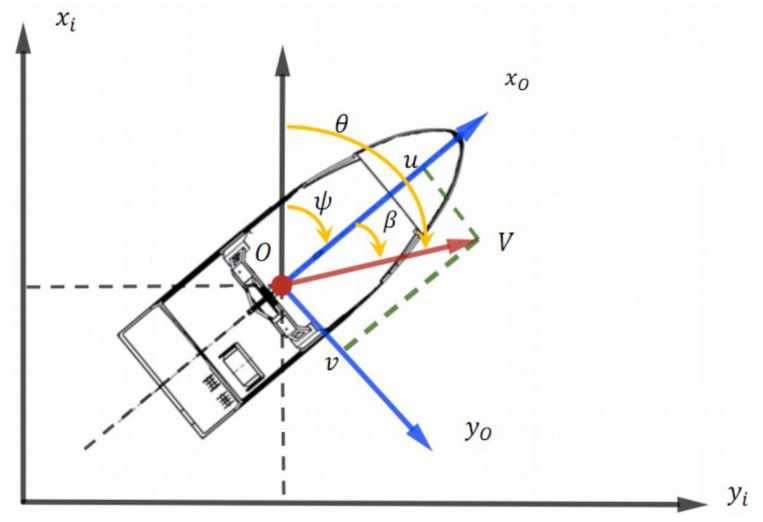
Schematic description of course, heading, and sideslip angles.

**Figure 2 sensors-23-04567-f002:**
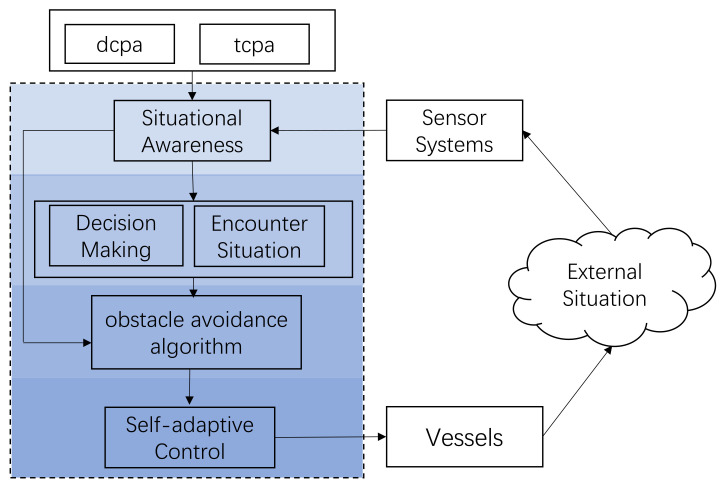
Architecture of intelligent vessels.

**Figure 3 sensors-23-04567-f003:**
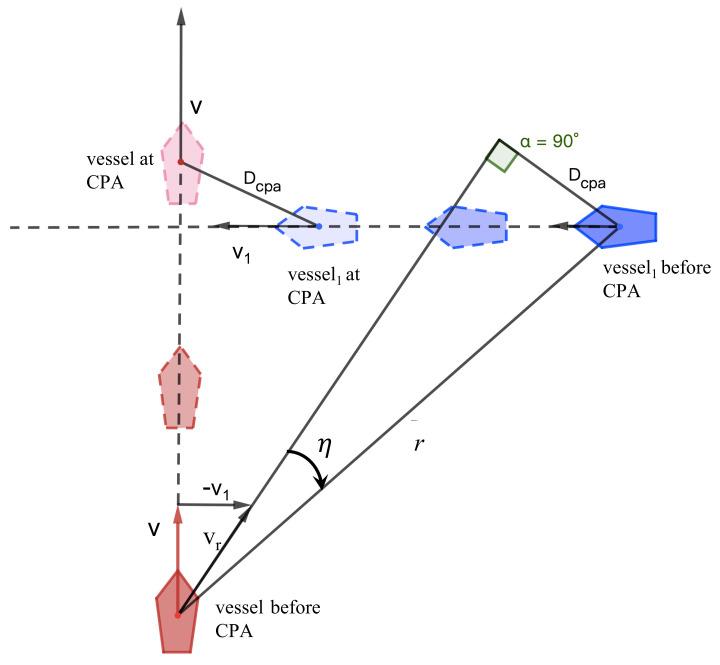
Distance at the closest point of approach.

**Figure 4 sensors-23-04567-f004:**
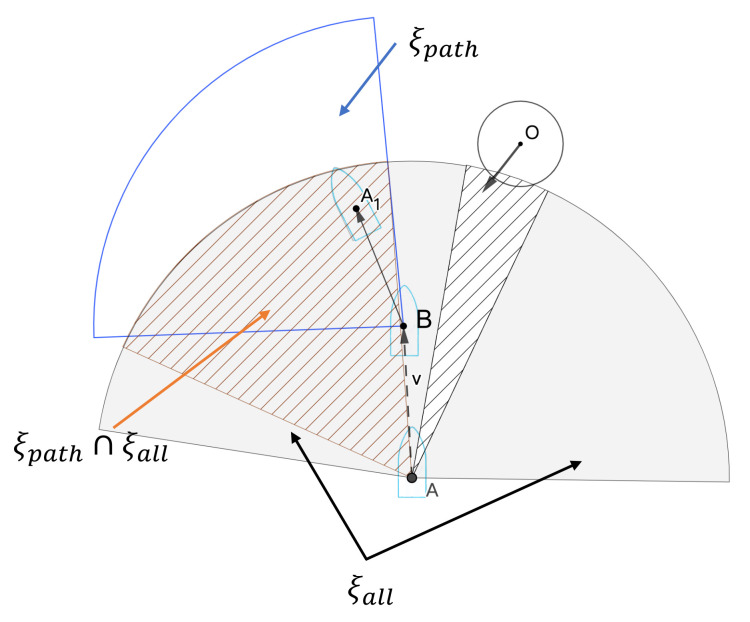
Velocity window sampling diagram.

**Figure 5 sensors-23-04567-f005:**
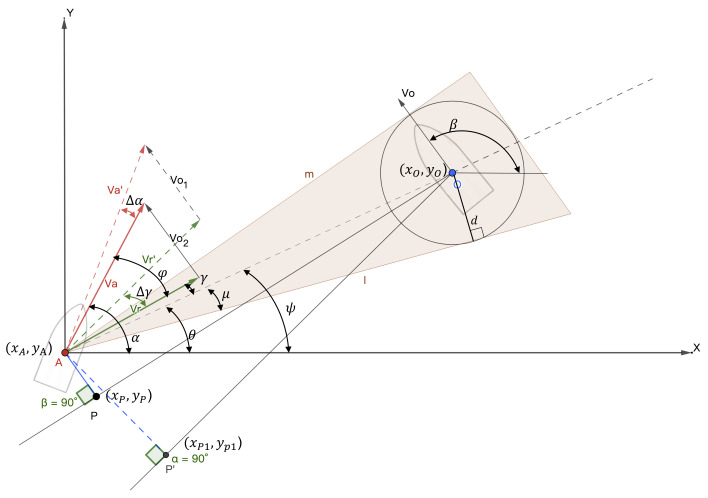
Improved velocity obstacle method and $d_{cpa}$.

**Figure 6 sensors-23-04567-f006:**
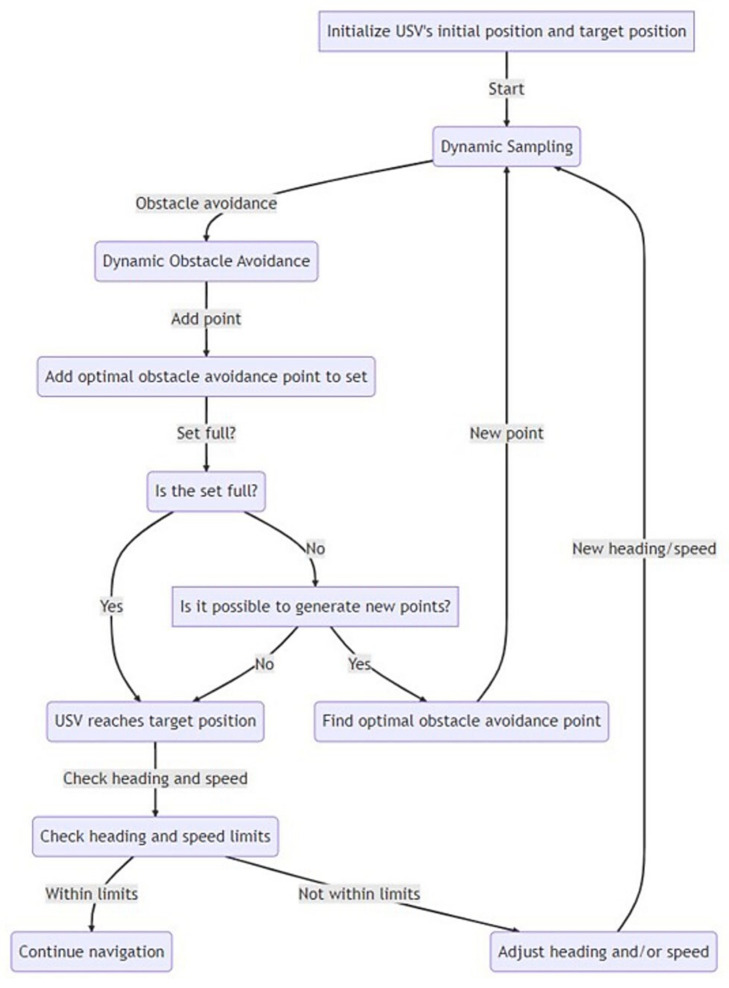
OCAP algorithm framework.

**Figure 7 sensors-23-04567-f007:**
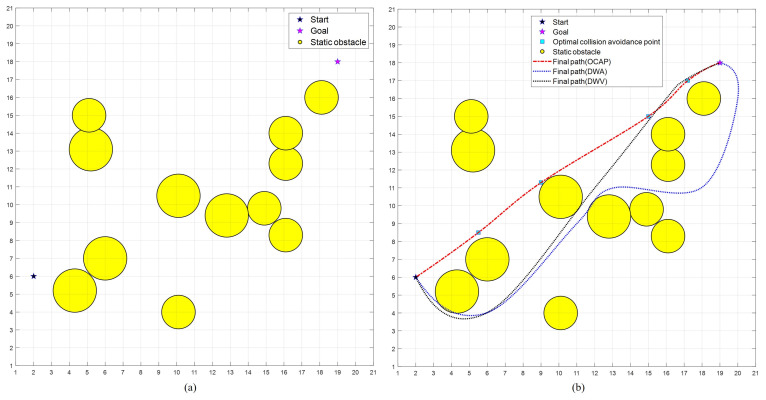
Simulation environment in Case1: (**a**) map at a static obstacle density of 0.04 and (**b**) comparison of paths generated by three optimization methods.

**Figure 8 sensors-23-04567-f008:**
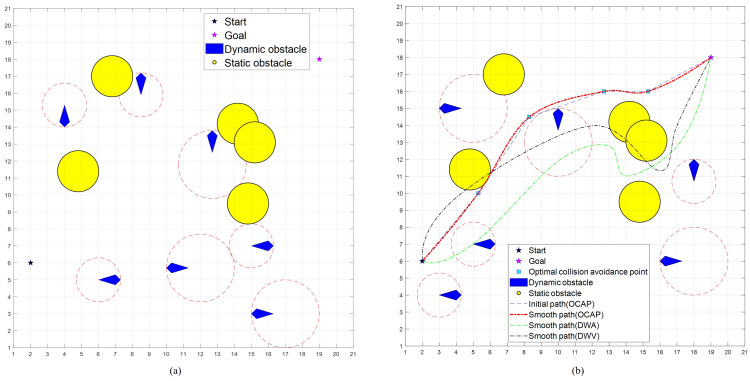
Simulation Environments in Case2: (**a**) random map (**b**) path generated by three algorithms.

**Figure 9 sensors-23-04567-f009:**
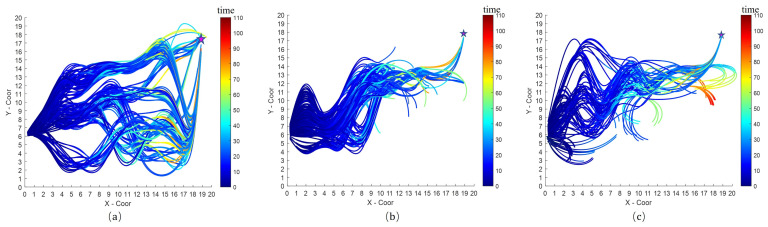
Simulation results of three optimization methods in Case2: (**a**) OCAP, (**b**) DWA, (**c**) DWV.

**Figure 10 sensors-23-04567-f010:**
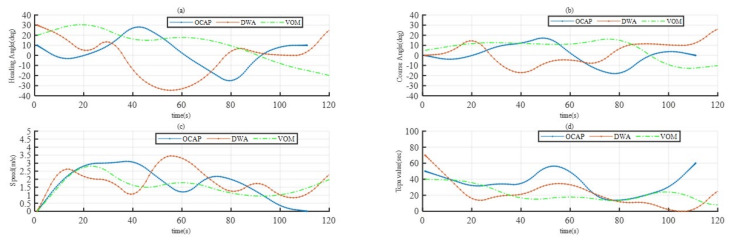
Three algorithm simulation parameters compared in Case2: (**a**) heading angle, (**b**) course angle, (**c**) USV speed to collision, (**d**) USV $t_{cpa}$.

**Figure 11 sensors-23-04567-f011:**
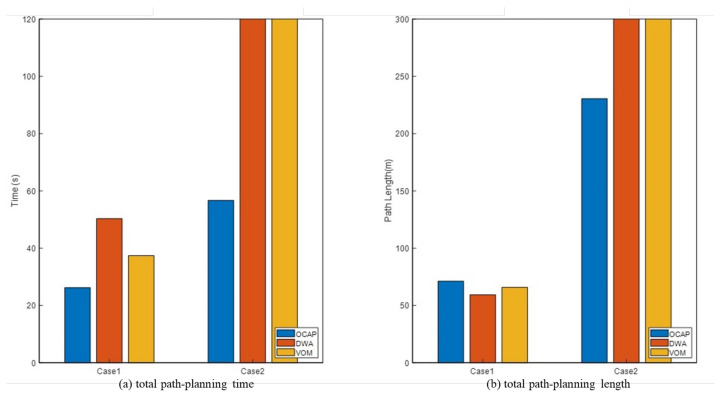
Three algorithms in optimal path-planning performance: (**a**) total path-planning time (**b**) total path-planning length.

**Table 1 sensors-23-04567-t001:** Simulation setup.

Case	Obstacle Position	Obstacle Radius	Obstacle Type	Obstacle Velocity
Case1	Constant	1, 1.3	Constant	-
Case2	Random	Random	Random	[0 0.3]

**Table 2 sensors-23-04567-t002:** Simulation results in Case1.

Case	Method	Success (%)	Time (s)	TL (m)	$\lambda_{\mathrm{dis}}$ (%)
1	DWA	100	5.377	19.521	9.04
1 time	DWV	100	3.502	15.804	9.92
	OCAP	100	3.268	14.287	4.87

**Table 3 sensors-23-04567-t003:** Simulation results in Case2.

Case	Method	Success (%)	Time (sec)	TL (m)	$\lambda_{\mathrm{dis}}$ (%)
2	DWA	60.98	65.377	29.521	25.04
100 times	DWV	58.42	57.502	25.804	25.92
	OCAP	82.73	60.268	24.287	18.09

**Table 4 sensors-23-04567-t004:** Comparison of OCAP and conventional algorithms at different obstacle densities.

Obstacle Density (%)	Method	Success	T (sec)	TL (m)	$\lambda_{\mathrm{dis}}$ (%)
	DWV	92.340	3.157	28.496	5.281
0.02	DWA	94.554	2.976	25.765	5.094
	OCAP	98.231	2.800	17.027	5.090
	DWV	12.340	81.653	48.064	33.317
0.06	DWA	34.554	88.910	45.109	40.338
	OCAP	80.231	79.772	37.407	26.060
	DWV	10.630	156.157	98.496	53.281
0.08	DWA	13.800	182.976	95.765	52.094
	OCAP	67.781	106.278	87.027	20.122

## Data Availability

Not applicable.

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
