# Peer review of "Optimized Dynamic Collision Avoidance Algorithm for USV Path Planning"

_sensors, 2023, doi:10.3390/s23094567_

Round 1

Reviewer 1 Report (New Reviewer)

The manuscript provides an optimal algorithm for collision avoidance for USV.  In this reviewer's opinion, the obstacle avoidance problem for USV is quite similar to that for UAV in many ways.  Thus, the authors should give the comparisons and differences between the development of an optimal collision avoidance algorithm for USV and that for UAV. Moreover, the authors should point out the difficulties for the development of the proposed algorithm. 

In (2) and (8), some parameters are not defined. 

The content of Sec. 3.2.2 looks quite complicated.  Use Fig.5 to give more geometrical explanations and intuitions on the derivation of  (13)-(19). 

Explain clearly how (20)-(24) are derived and what are their physical meanings. 

Author Response

Dear reviewers, please refer to the attached document.

Reviewer 2 Report (New Reviewer)

Dear Authors,
Thank you for your work on collision avoidance between ships and others obstacles. Your algorithm considers a scenario unrealizable because it considers a complete knowledge of the actors (other vessels and obstructions in general) present in the scenario.

The simulation seems like a videogame where the physics of the controlled ship is very simplified. It is not acceptable.

From a formal point of the developed algorithms, you do not consider the algorithm complexity analysis (e.g. Cormen O-notation).

Please, provide a formal specification of the simulation.

Please provide a deep analysis of your idea to get the paper accepted.

Regards

Author Response

Dear Reviewer,

Please find attached the revised version of our paper. 

Round 2

Reviewer 1 Report (New Reviewer)

The authors have revised the manuscript in accordance with my comments. The manuscript is accepted for publication.

Reviewer 2 Report (New Reviewer)

Dear Authors,
Thank you for following my suggestions. The changes you have made permit me to accept the paper.

Regards

This manuscript is a resubmission of an earlier submission. The following is a list of the peer review reports and author responses from that submission.

Round 1

Reviewer 1 Report

This paper discusses a subject applicable to ship collision avoidance, which is of great significance for improving the navigation safety of USV. In my opinion, the paper is considered a valuable and interesting study in the related field. However, as there is always room for improvement, the reviewer comments are as follows.

1. Abstract should be concisely and clearly described, including the background, purpose, method, result, and conclusion of the study.

2. In the description, ambiguous expressions should be avoided and quantitative numerical values or objective grounds should be presented. For example, it is not appropriate to directly express the ship's speed as high speed and low speed in Table 1. How to determine the specific evaluation standard and definition range should be explained.

3. It is mentioned in the abstract that this study satisfies the rules of collision avoidance at sea, However, there seems to be no relevant content mentioned in the article. It is suggested to explain it in the section on experimental results. In addition, The rule refers to International Regulations for Preventing Collisions at Sea (COLREGS). Please check whether the expression of relevant professional terms is reasonable.

4. In the conclusion of section 5, it is necessary to describe the limitations of this study and future research, and it is suggested to add relevant content.

5. The format of references should follow the corresponding standards to ensure accuracy and unity. Please carefully check and modify.

6. In line 243, I think there is a problem with the punctuation after the word “respectively”. Please check the rest of the text.

7. Please carefully check whether the description of relevant professional terms in the full text is correct and uniform. For example, the expression of the ship in line 166, line 174 and Figure 4 is Ship, vessel, and vehicle respectively.

8. In Section 4, whether the test results are contingent, and whether the selection of initial values, obstacle positions and target positions will affect the experimental results. Model 2 is not applicable in a low speed environment, but it is applicable in the telling situation, whether it is a success of chance or not. Furthermore, it is not reasonable to assume that a scenario has a 0% or 100% success rate based on its experimental results. It is suggested to make a more reasonable description of the conclusion.

Reviewer 2 Report

This paper proposes an improved optimal collision avoidance point model (DWOV) for USV. The authors have done a lot of work. But there are several problems.

1. According to the title, this paper aims to study the optimized dynamic collision avoidance algorithm, but many parts of this paper describe the content of path planning. In particular, the conclusion section points out that the method proposed in this paper is "a new local path planning method". Although the “collision avoidance” and “path planning” are related, it is suggested that the authors focus on the key research contents of this paper so that readers can better understand this paper.

2. In this paper, there is a lack of in-depth analysis and summary of the latest related researches on USV collision avoidance.

3. It is suggested that the authors highlight the innovations and contributions of this paper in the abstract and introduction section.

4. About the proposed “improved DWA algorithm”, “improved Velocity Obstacle algorithm” and “Improved Dynamic Window Approach” in this paper, please clarify the specific improvements in the corresponding sections of the paper.

5. As mentioned in line 193, in this paper, the USV is regarded as a give-way vessel and the target ship is regarded as a stand-on vessel. This assumption is too idealistic, and the actual situations should be more complicated.

6. The “velocity barrier method”, “Obstacle Velocity Method (OVM)” and VOM used in the paper are inconsistent and need to be corrected.

7. There are some grammatical errors in the paper. 

Reviewer 3 Report

Thank you for your contribution that covers an actual research topic.

In general the paper provides a detailed introduction to the problem space with numerous citations and explainationes.

The core of the paper needs improvement. The DWA ist very old and designed 40 years ago. The body of work ist quoted sufficiently and it not gets clear what ist the new contribution / add on to the existing research.

The description of the approach is weak and has not the same level as the introduction.

The applied Nomoto Model is very  simple.  Its not clear which model is used in the simulation. Any details of the applied dynamic model is missing . Therefore the evaluation is not sufficient to prove the quality of the concept. 

Round 2

Reviewer 2 Report

The authors have improved the manuscript. However, there are still some typing errors that need to be corrected.

Author Response

Thank you very much for your correction, there are some punctuation marks in the paper that were not entered in the correct input method.

In line 3, we have added a space between "collision"and"called",

In line 375, we have deleted  the redundant "the".
All initial letters of words after semicolons have been corrected to lower case.

In line 50, we have corrected "The" to "the". 

Reviewer 3 Report

Thank you for the revised manuscript. I am sory to recognize that the main obstactles of the paper are not addressed (e.g.  reference to existing reseach, the model in the evaluation etc).

Author Response

We appreciate this comment from the reviewer and realize that these differences may not have been expressed clearly enough in the previous manuscript. We have improved the original manuscript, particularly in the introduction (paragraph 3) and discussion (paragraph 1), to more clearly illustrate these benefits of DWOV algorithm.